# Functional Bacterial Amyloids: Understanding Fibrillation, Regulating Biofilm Fibril Formation and Organizing Surface Assemblies

**DOI:** 10.3390/molecules27134080

**Published:** 2022-06-24

**Authors:** Thorbjørn Vincent Sønderby, Zahra Najarzadeh, Daniel Erik Otzen

**Affiliations:** 1Interdisciplinary Nanoscience Center (iNANO), Aarhus University, Gustav Wieds Vej 14, 8000 Aarhus, Denmark; thorbjoern.soenderby@gmail.com (T.V.S.); zahra_najarzadeh@yahoo.com (Z.N.); 2Sino-Danish Center (SDC), Eastern Yanqihu Campus, University of Chinese Academy of Sciences, 380 Huaibeizhuang, Huairou District, Beijing 101400, China

**Keywords:** bacterial amyloid, biofilm, curli, FapC, imperfect repeats

## Abstract

Functional amyloid is produced by many organisms but is particularly well understood in bacteria, where proteins such as CsgA (*E. coli*) and FapC (*Pseudomonas*) are assembled as functional bacterial amyloid (FuBA) on the cell surface in a carefully optimized process. Besides a host of helper proteins, FuBA formation is aided by multiple imperfect repeats which stabilize amyloid and streamline the aggregation mechanism to a fast-track assembly dominated by primary nucleation. These repeats, which are found in variable numbers in Pseudomonas, are most likely the structural core of the fibrils, though we still lack experimental data to determine whether the repeats give rise to β-helix structures via stacked β-hairpins (highly likely for CsgA) or more complicated arrangements (possibly the case for FapC). The response of FuBA fibrillation to denaturants suggests that nucleation and elongation involve equal amounts of folding, but protein chaperones preferentially target nucleation for effective inhibition. Smart peptides can be designed based on these imperfect repeats and modified with various flanking sequences to divert aggregation to less stable structures, leading to a reduction in biofilm formation. Small molecules such as EGCG can also divert FuBA to less organized structures, such as partially-folded oligomeric species, with the same detrimental effect on biofilm. Finally, the strong tendency of FuBA to self-assemble can lead to the formation of very regular two-dimensional amyloid films on structured surfaces such as graphite, which strongly implies future use in biosensors or other nanobiomaterials. In summary, the properties of functional amyloid are a much-needed corrective to the unfortunate association of amyloid with neurodegenerative disease and a testimony to nature’s ability to get the best out of a protein fold.

## 1. Dedication to Sir Chris Dobson by Daniel Otzen

Chris was a luminary in his field—he lit up the literature on amyloid proteins like a lighthouse, shining bright insight and knowledge into many different aspects of this part of the protein universe. Aside from the prodigious outpouring of original research from him and his group, not to mention his always inspiring and witty lectures (which were never the same, despite his frequent stage appearances), the scientific landscape is dotted with his incisive (and often single-author) reviews that appeared at a breathtaking pace over several decades. For me, one of the real eye-openers was his 2003 review in Nature [1], simply titled “Protein folding and misfolding,” which I read as soon as it came out. I leant heavily on it to prepare a lecture for high-school teachers as part of a Danish “*life-long learning*” initiative to keep these teachers informed about the latest developments in their fields. It struck me as very appropriate to consult Chris in such a situation. Although I was reasonably informed about the amyloid field at this stage, Chris’ magisterial overview revealed many new aspects for me. One sentence in particular caught my attention: “Nevertheless, there is increasing evidence that the unique properties of amyloid structures have been exploited by some species, including bacteria, fungi and even mammals, for specific (and carefully regulated) purposes.”. This brought my hitherto unenlightened attention to the world of functional amyloid, which had important consequences for me. It introduced me to wonderful colleagues such as Matt Chapman (whose seminal work on curli and CsgA was referenced by Chris in the above quote) and inspired me, along with fellow colleagues such as Per Halkjær Nielsen and Morten Dueholm at my host institute (at the time Aalborg University), to embark on a journey into the world of useful aggregation. We started by showing that amyloid was all around us in the bacterial world [2,3] and then went on to focus on some of those systems in more detail. That journey is still ongoing almost two decades later, and we will use this opportunity to tell a little about our research in that area in this paper. Thank you, Chris, for that tip!

## 2. The Conundrum of Protein Misfolding and Aggregation

Amyloids are highly organized, β-sheet-rich protein aggregates [4]. They can form from a wide variety of different proteins, suggesting—as Chris Dobson vigorously and convincingly emphasized—that proteins have a *generic* ability to form amyloid in addition to their tendency to assume globular folds [4,5]. Indeed, in some cases (again as shown by Chris and colleagues [5]), the amyloid state is more thermodynamically favorable than the native state (Figure 1a). Fortunately, that does not mean that life automatically degenerates to amyloid soup (although it may have started this way) [6]. This is due to several protective mechanisms. One of these is the cooperative nature of protein folding [7]. As a result, only a miniscule fraction of the protein population will be unfolded at any given time, and most aggregation-prone regions remain safely tucked away into the protein interior [7]. Therefore, proteins need to cross a high activation barrier to unfold and reach aggregation-prone states. On top of that, they need to encounter and associate with other similarly conformationally inclined molecules to start the aggregation reaction. These aspects combine to make amyloid formation a kinetic, rather than a thermodynamic, challenge [5]. In addition, cells contain protective mechanisms such as folding chaperones, which aid protein folding, e.g., by screening aggregation-prone regions during folding and sequestering proteins in compartments until they are safely folded [7]. Further, degradation mechanisms quickly remove misfolded proteins in normally functioning cells [7]. Nevertheless, destabilization of the native state (or stabilization of the denatured state) can reduce the kinetic barrier between the folded and denatured state, leading to more frequent traffic between the two states, leading to greater exposure of aggregation-prone regions [4]. The barrier to unfolding is completely removed in intrinsically disordered proteins (IDPs), which are largely disordered and only contain transient structure. For them, the only remaining obstacle is the suite of intermolecular contacts and rearrangements that need to be made to start the process of organized aggregation to the amyloid state [4] (Figure 1b).

## 3. Amyloids in Sickness and in Health

It follows from the above considerations that unwanted accumulation of amyloid in vivo will be favored by either protein destabilization or the involvement of IDPs. Examples of the former include the protein transthyretin, a tetramer which upon mutagenic destabilization can dissociate and misfold to amyloid structures, leading to cardiac amyloidosis [8]. Stabilization of the tetramer should in principle slow down or even prevent disease progression. Indeed, the drug tafamidis (which binds to two otherwise empty ligand-binding sites on the tetramer) does exactly that [9,10], making it the first and so far only example of a clinically approved drug that specifically prevents amyloid formation. Unfortunately, most amyloid-associated diseases involve proteins or peptides that are IDPs [4], e.g., Parkinson’s (α-synuclein), Alzheimer’s disease (amyloid-β and tau) and type-II diabetes (Islet amyloid polypeptide) [4,11,12,13]. They do not lend themselves to such straightforward stabilizing ligand strategies as using tafamidis but require more complicated approaches that somehow target the unwanted associated species without compromising their (more) innocent precursors.

These misfolding diseases should not prevent us from appreciating some innately striking features of amyloids: their stability and simplicity [14]. Their structure only requires copies of a single type of protein which can rapidly stack up on top of each other to form long fibrils [15]. Why not use these properties for something useful? Indeed, many organisms use amyloids for beneficial purposes, and so-called **functional amyloids** (usually based on IDPs) are widespread among organisms and even exist in humans [16,17,18,19,20]. Bacteria in particular have been very active in their exploitation of these self-assemblies [19,20,21], which amongst other things can be used to strengthen the biofilm in which most bacteria are embedded [19,20,22]. Although useful for the bacteria, biofilms are generally a nuisance for humans, as they protect bacteria against physical, chemical and biological attacks. This leads to increased antibiotic resistance of bacteria on surfaces such as wounds and medical implants [23], but also in more innocuous settings, such as clothes, contact lenses, water flow systems and even oil pipes [24]. Thus, even when an amyloid plays a useful role (for some), there is strong motivation (for others) to prevent it from forming. Nevertheless, their formation and structure are highly instructive and provide an excellent showcase of how to control and benefit from a potentially proteotoxic fold. As we will describe in greater detail below, the best-studied functional amyloids (those in *E. coli* and *Pseudomonas*) are organized as multiple copies of relatively simple repeats, which lack structure in the monomeric state but fold at the multimeric level in a simple but robust framework thanks to a large number of stabilizing inter- and intramolecular contacts, which are built up in a hierarchical manner. This is in stark contrast to pathological amyloid proteins where there are no repeat sequences to provide an orderly framework for assembly; consequently, many different aggregate types (polymorphs) exist which can form in different ways and typically involve cytotoxic partially-folded intermediate states.

## 4. Functional Bacterial Amyloids

Matt Chapman and colleagues reported the first example of functional bacterial amyloid (FuBA), namely, curli in *Escherichia coli* (*E. coli*) [20], hence giving rise to the term *curliated E. coli*. Since this discovery in 2002, numerous other FuBAs have been discovered, including functional amyloid in *Pseudomonas* (Fap) [19,25]. The curli and Fap systems remain the most well-characterized FuBA systems, and we will briefly introduce these two systems.

## 5. The Curli System

The curli system is very widespread, occurring in several distinct phyla within the bacterial kingdom [26]. Curli fibers, which extend into the extracellular matrix of *E. coli*, are involved in biofilm formation and play a role in surface colonization and host-tissue contact [20,27]. Curli fibers consist of two proteins, CsgA and CsgB, in vivo [20], of which CsgA constitutes the vast majority [28]. While purified and isolated CsgA very readily and robustly forms fibrils on its own in vitro, the nucleator protein, CsgB, is essential for correct CsgA aggregation in vivo [29]. The biogenesis of curli is tightly regulated and involves five other proteins besides CsgA and CsgB. These are encoded from two operons that are divergently linked, *csgBAC* and *csgDEFG* (Figure 2) [30] but which collaborate in beautiful concord to orchestrate the carefully controlled production of curli projecting from the surface of *E. coli*’s outer membrane. The transcription regulator, CsgD, activates the expression of the curli subunits, CsgA and CsgB, from the *csgBAC* operon, in addition to the periplasmic chaperone, CsgC [30]. The Sec export pathway is responsible for the transport of all curli proteins across the inner membrane to the periplasmic space (except the transcription activator, CsgD) [31]. Once in the periplasmic space, CsgA, CsgB, and CsgF are further transported across the outer membrane through the outer membrane pore-forming protein, CsgG [31]. CsgE forms a complex with the pore-forming protein, CsgG, and functions as a gatekeeper for the secretion of CsgA and CsgB through CsgG by recognizing outer-membrane signal sequences in CsgA and CsgB [32]. The nucleator CsgB is transported to the cell surface, where it is attached to the outer membrane, stabilized by contact with CsgF (Figure 2) [33]. CsgB is homologous to CsgA and presents a binding surface for CsgA, which is secreted from the cell as an IDP, but fibrillates rapidly once it makes contact with CsgB [29,34]. CsgC is a very efficient amyloid inhibitor/chaperone which binds CsgA and CsgB during transport in the periplasmic space and maintains them in the monomeric state, thereby inhibiting premature fibrillation of CsgA and CsgB [35,36].

## 6. The Fap System

The curli and Fap systems are evolutionarily unrelated, yet their biogenesis is remarkably similar [19,20]. The Fap system is especially abundant within the Proteobacteria phylum [37,38] and may contribute to the virulence of the pathological strain *Pseudomonas aeruginosa* (*P. aeruginosa*) [19,39], thanks to its ability to stabilize the biofilm mechanically [40]. Like curli, Fap biogenesis requires several accessory proteins (FapA–F) whose roles appear to be similar to those in the curli system [41]. However, there are differences all the same. Unlike curli, Fap proteins are transcribed from a single operon (Figure 2) [19]. FapC is the main structural component of Fap fibrils, but they also contain smaller amounts of FapB and FapE [19,41]. FapB is predicted to be a nucleator protein analogous to CsgB [19,41]. The role of FapE is less clear but it was found to be essential for Fap secretion [42]. The roles of FapA and FapD are unclear, but they are likely to be chaperones [19,41,42]. Curiously, deletion of FapA leads to Fap fibrils completely made of FapB [41]. Like the curli proteins, the Fap proteins utilize the Sec transport pathway for transport across the inner membrane [42]. Additionally, as in the curli system, a dedicated outer membrane pore protein (FapF) allows transportation of FapB, FapC and FapE across the outer membrane [19,42]. However, FapF and CsgG are completely unrelated. FapF is a trimeric porin, each subunit of which appears to be able to work as an independent channel [42], whereas CsgG is a 9-mer, and the individual subunits each contribute to the formation of a single channel [43].

Insets of amyloid structures are from [38,44]. The solved structures of CsgC and CsgE have been deposited in the PDB with PDB IDs 2Y2Y and 2NA4, respectively.

## 7. The Major Curli Component, CsgA

The complicated and elegantly regulated biogenesis of curli and Fap illustrates that these systems are evolutionarily top-tuned fibrillation machineries. This becomes even clearer when we inspect the predicted structures of the major amyloid components, CsgA and FapC, both of which contain multiple imperfect repeats.

## 8. Predicting FuBA Structure Is Easier Than Predicting Pathological Amyloid Structure

The difficulties in solving amyloid structure experimentally (e.g., due to their insoluble and non-crystalline nature), have motivated the use of computational methods to predict the structures of functional amyloids [30,38,44,45,46,47,48]. This has been applied particularly to CsgA [30,44], the curli nucleator protein, CsgB [45,46] and FapC [38]. These amyloid structure predictions have primarily depended on multiple sequence alignment (MSA) to detect patterns of co-evolutionary residue pairs, which often indicates residue-residue contacts in the folded protein [49]. This method has appeared to work well for functional bacterial amyloids, which are functional in their fibril state and are evolutionarily widespread, allowing MSA of many homologous proteins [30,38,44,45,46,47,48]. In addition, MSA has provided insight into the polarity of bactofilin filaments from *Thermus thermophilus*, which complimented cryo-electron microscopy data of the bactofilin filaments [48]. In contrast, MSA has little relevance for pathological amyloids, whose aggregation results from misfolding rather than evolutionary optimization [4] and often results in amyloids with a significant degree of polymorphism and heterogeneity compared to functional amyloids [50]. The newly released AlphaFold2 protein structure predictor is also based on co-evolutionary MSA analysis, coupled with the training of a neural network based on protein structures from the PDB [47,51]. AlphaFold2 is a promising tool for predicting structures of functional amyloid [52]. FuBAs are an interesting test of the capabilities of AlphaFold, because they are IDPs that gain structure upon association, but in contrast to pathological amyloids, their final structure is the result of evolutionary pressure.

Evolution has also ensured that CsgA consists of five 19–23 residue-long imperfect repeat sequences preceded by a 22-residue N22 sequence (Figure 3a,b) [30,44]. The overall length of mature CsgA is 131 residues (after cleavage of a 20-residue inner membrane signal sequence) [20]. All five repeats contain a Ser–Xxx_5_–Gln–Xxx_4_–Asn–Xxx–Ala–Xxx_3_–Gln consensus sequence (Figure 3b) [53,54]. Each repeat sequence of CsgA is predicted to form a hairpin motif consisting of two β-strands linked by a β-turn (Figure 3a) [30,44]. These hairpins are stacked on top of each other, so that each strand engages with the other hairpin strands, forming two parallel β-sheets side by side. The repeats are internally connected by short (4–5 residue) loop sequences [44,55]. This arrangement results in a β-helix structure [30,44,45,46,47]. Several studies have predicted similar β-helix structures of CsgA based on in silico covariance analysis and molecular dynamics [30,44,45,46,47] which are consistent with (admittedly limited) solid state NMR data [56]. However, the in silico predictions cannot distinguish between a left-handed or right-handed β-helix, since they are equally stable [44]. Frustratingly, CsgA structure predictions deposited in the AlphaFold structure database also include both left- and right-handed β-helix predictions [47,51]. While the AlphaFold predictions of CsgA have not been submitted to peer review and should be interpreted with caution, they are very similar to the previously peer-reviewed predictions [30,44,45,46,47]. Despite these ambiguities, all in silico predictions of CsgA structures consist of multiple parallel ladders resulting from the vertical alignment of conserved residues. A hydrophobic core formed from “ladders” of L/I/M/V/A residues is flanked by ladders of conserved Ser, Gln and Asn residues, which are predicted to form hydrogen-bond networks (Figure 3c) [30,44,45,46,47].

Although the five repeat sequences of CsgA are similar, they differ from each other in important ways. Repeats R2–R4 contain gatekeeper Asp and Gly residues which, given their higher polarity, reduce the aggregation propensity of CsgA (Figure 3b) [57]. They are called gatekeepers because their mutation to the corresponding residues found in R1 and R5 (Asp/Gly → Asn/Ser/Leu/His) accelerates fibrillation markedly compared to wild-type CsgA [57]; i.e., they serve to block (or reduce access to) certain types of interactions. They do so by disrupting a sequence of residues with a certain physical-chemical profile, just as we proposed for the amyloidogenic Aβ peptide [58]. Ironically, the more aggressive aggregation of the gatekeeper-less CsgA mutant compromises bacterial viability and make fibrillation independent of the helper proteins CsgB and CsgF, in vivo [57]. These observations underline an important point: too vigorous fibrillation is to be avoided because it threatens efficient control over the fibrillation process [57]. Perhaps unsurprisingly, the two terminal repeats R1 and R5, which constitute the two flanks of each CsgA monomer, are the most important repeats for driving the fibrillation of CsgA [59]. Exchanging either R1 or R5 with R3, resulting in the mutants R32345 or R12343, drastically reduces curli formation [59]. On the other hand, R1 and R5 are interchangeable, since R12341 and R52345 both form curli fibrils that are indistinguishable from wild-type fibrils [59]. All this is corroborated at the molecular level when the repeat sequences are synthesized as individual peptides and allowed to fibrillate. The peptides fibrillate in the ranking R5 > R1 > R3, whereas R2 and R4 do not fibrillate at all [53,59].

## 9. The Major Fap Component FapC: A Flexible Number of Imperfect Repeats and Variable Linker Lengths

Like CsgA, FapC contains imperfect repeats [19]. FapC is a considerably longer protein (250–340 residues, depending on the species [19]) than CsgA (131 residues), but only three imperfect repeats are found in FapC, as opposed to five in CsgA [19]. In *Pseudomonas* sp. UK4 (where we originally discovered FapC [19]), the three repeat sequences of FapC are each about 34 residues long, and are connected by two linker sequences of 31 and 39 residues, respectively [19]. These sequences are clearly much longer than the tight turns found in CsgA. The Fap operon is found within three classes of the phylum Proteobacteria, where FapC is found to vary considerably in length due to variations both in linker length (70 in *Pseudomonas* sp. UK4 but 160 in *Pseudomonas aeruginosa*) and in the number of imperfect repeats (three in most species but up to 16 in *Vibrio*) [37]. Nevertheless, the same combination of covariance-derived restraints and molecular dynamics used for CsgA also led to a predicted β-helix structure for FapC in which each repeat forms a β-hairpin (Figure 4a) [38].

Like CsgA, the imperfect repeats of FapC contain conserved Ser, Ala, Gln and Asn residues (Figure 4c) [19]. The longer repeat sequences and linkers in FapC relative to CsgA [19,20] indicate a less streamlined design could be attributed to the Fap system’s evolutionary youthfulness compared to the curli system. That is, FapC has not yet had time to lose some of its unordered parts protruding from its lateral regions (Figure 4b) [37]. However, there are also other possibilities. Structural prediction of FapC by AlphaFold results in a markedly different type of structure which integrates the entire sequence into a highly structured fold (Figure 4d). Here, both the predicted repeat sequences, the linkers and the N-terminal sequence form stacked β-strands in the core of protein. In this predicted structure, the N-terminal and the linkers L1 and L2 stack on top of each other and form “ladders” of conserved residues, just like the repeat sequences (Figure 4e). There is a good deal of evidence that the repeat sequences are the main driving forces of the amyloidogenicity of FapC [19,60,61]. Systematic removal of repeat sequences from FapC has two important consequences: it makes FapC more vulnerable to fragmentation during fibrillation [61] and it reduces the stability of mature FapC fibrils [60]. FuBA stability is here quantified as resistance to dissolution in formic acid [60], one of the few solvents able to reduce FapC to monomers that are visible on SDS-PAGE [19,20]. Formic acid has a pK_a_ of 3.75 so is not a particularly strong acid [62], but the high concentration needed to dissolve FuBA (~17 M formic acid is needed to dissolve 50% of wildtype FapC [60]) leads to a combination of low pH (<2) and a high concentration of organic solvent, which seems to do the trick [60].

Remarkably, FapC is able to fibrillate even after the complete removal of repeat sequences (although the fibrils are enormously destabilized against formic acid), indicating that the linker regions of FapC play a role in fibrillation (or at least step in as amyloid substituents if the main drivers are removed) [61,63,64]. This is supported by our study on the effect of denaturants on FuBA fibrillation [63], which suggested paradoxically that removal of FapC repeats *increases* the amount of buried surface area per protein molecule. That is, that the number of residues in the fibril backbone *increased* once we removed two or three repeat sequences [63]. The best explanation for this observation is that the linker regions step in to become part of the fibril backbone, once the “proper” amyloid components (repeat sequences) are removed. Currently this remains speculative; however, AlphaFold predictions of a series of different deletions of FapC show quite dramatic changes in structure once two or three repeats are removed (Figure 5), vindicating the observed dramatic changes in stability [61]. Interestingly, in computational disorder predictions of FapC, the repeat sequences obtained higher scores of disorder propensity compared to the linker regions [63]. This is in good agreement with previous systematic analysis which showed a clear relationship between intrinsic disorder and amyloid propensity [61]. The hypothetical structure of a FapC homologue from *Acidothiobacillus* with eight imperfect repeats assigns one “layer” to each repeat in an 8-layer β-helix (Figure 5). Clearly, it will be very exciting to see the experimental structure of FapC once it is hopefully revealed by CryoEM.

There is an important biological lesson in these observations which reinforces the conclusions from gatekeeper residues in CsgA. Uncontrolled aggregation through secondary processes, such as fragmentation or nucleation along the sides of the fibrils, is undesirable because it can lead to runaway processes which accelerate exponentially (i.e., go from baseline to saturation in a very short time). In contrast, primary processes (nucleation followed by elongation) occur at a much more moderate rate, leading to a steady increase in fibril growth over a longer time course which is compatible with the time scales of bacterial growth and biofilm consolidation [66]. Indeed, there is good evidence that primary processes dominate in benign aggregation and secondary processes in pathological ditto (G. Meisl et al. and T.P.J. Knowles, unpublished observations). This may rationalize why FuBAs, under normal conditions, are not cytotoxic to their host organisms [57], and provides more insight into the cytotoxicity of pathological amyloids in human diseases [67,68].

## 10. Probing FuBA Fibrillation as Folding Steps with Denaturants: Similar Folding during Nucleation and Elongation Activation Steps

The fact that CsgA and FapC are evolutionarily unrelated yet remarkably similar gives us a unique chance to better understand the “design” of amyloid. How do unrelated amyloids converge to relatively similar design blueprints? [63]. Monomeric CsgA is very unstructured according to hydrogen-deuterium exchange [69], light scattering, single fiber AFM-analysis [70], secondary structure determination [71] and NMR studies on the pre-amyloid state [72] and MD simulations of peptide segments of CsgA [73]. However, upon association, CsgA folds into a uniform amyloid state in what appears to be a simple two-state model [69,70]. FapC appears to follow the same model [74]. Thus, the aggregation of both proteins is analogous to the folding of globular proteins, and should therefore be subject to the same type of scrutiny, e.g., using denaturants such as urea and guanidinium chloride. Titration of proteins with increasing concentrations of denaturant typically leads to a sigmoidal equilibrium unfolding curve [75], which is extensively used to study protein stability and folding [76]. Denaturants generally favor unfolded states by preferential binding, so the larger the surface area available for binding, the more the protein is stabilized. The free energy of unfolding depends linearly on denaturant concentration with a slope called the *m*-value, which is proportional to the increase in surface area associated with the unfolding step [77,78]. Thus, the *m*-value directly reports on the extent of folding. This applies both to equilibrium values and kinetics; in the latter case, *m*-values report on the difference in surface area between the ground state (e.g., the native state) and the transition state. Since denaturants decrease folding rates, they would also be expected to reduce aggregation rates—unless aggregation requires prior unfolding from a native state. Thus, denaturants reduce the rate of amyloid formation by the IDPs Aβ40 and Aβ42 (which do not need to unfold before aggregating) [79,80] but increase rates for aggregation of the globular protein insulin [81]. The situation is much less complex with FuBA: Urea reduces both the nucleation rate and the elongation rate of both CsgA and FapC [63], and *m*-values for nucleation and elongation are very similar, suggesting a similar degree of structural consolidation (folding, i.e., burial of surface area) in nucleation and elongation. That is, elongation of FuBA is a natural extension of nucleation. Interestingly, for the FapC mutant lacking all three imperfect repeats, we see an increase in *m*-values, suggesting that more of the protein sequence is integrated into the amyloid core, which again is consistent with the concept of “stand-by” amyloidogenic sequences stepping up to form fibrillar regions once the core amyloidogenic repeats have been removed (Figure 6).

The simplicity of FuBA folding is revealed in other ways. In globular protein folding kinetics, a tell-tale sign of folding intermediates is the lack of linearity or roll-over at low denaturant concentrations in chevron plots (log folding rate constants vs. [denaturant]) [77,78]. Conversely, strict linearity implies simple two-state folding. Both CsgA and FapC show a log-linear relationship between elongation rate constants and [urea] between 0 and 8 M urea [63]. This rules out detectable intermediates during FuBA elongation, in contrast to Aβ1-40, which was recently proposed to go through a “frustrated” complex intermediate [82]. This is manifested as in increase in the elongation rate of Aβ up to 1.5 M urea [82], after which there is a decrease in the elongation rate, as described by other groups for both Aβ1-40 and Aβ1-42 [79,80]. For globular proteins, an increase in folding rate with [denaturant] at low [denaturant] is evidence of an off-pathway species which has to unfold back to the denatured state for folding to proceed [83,84]. This usually gives way to a more conventional decline in folding rates at higher [denaturant], so the off-pathway species is destabilized and does not accumulate. Similarly, the increase in elongation rate of Aβ at low [urea] can be seen as a destabilization of the “frustrated” (off-pathway) intermediate which leads to a “smoothing” of the energy landscape and causes less kinetic trapping in the intermediate state [82]. The apparent lack of any frustrated intermediate in FuBA elongation indicates that this is a “cleaner” two-state process with a single harmonious folding step [63].

Nevertheless, there is an important difference between the nucleation and elongation steps: molecular chaperones which inhibit fibrillation of CsgA and FapC primarily target the nucleation step [85] and not the elongation step. We reached this conclusion after analyzing four different human chaperones and the curli chaperone CsgC. Remarkably, the best chaperone was not CsgC but the human protein DNAJB6 which works at such low sub-stoichiometric ratios that it most likely blocks growth of the nucleus itself rather than the isolated monomer; this also explains its preference for nucleation rather than elongation. This common mode of action by completely unrelated chaperones is a strong reflection of the simplicity of bacterial amyloid formation, and primary nucleation is the main step generating new fibrils. At the same time, the fact that so many different chaperones with no evolutionary relationship to each other can target the same (optimized) aggregators suggests a commonality of anti-amyloid activities across nature, perhaps driven by the recognition of amyloidogenic hotspots within the primary structure. In support of this, CsgC can inhibit human αSN fibrillation [35,86] and reduces both CsgA and FapC fibrillation in vitro [36]. The specific molecular basis for chaperone action must remain speculative, though the efficiency of inhibition seems to scale with affinity of binding to the monomeric state [85]. In addition, we note that FapC is more resistant to chaperone action than CsgA, and this may reflect the presence of less amyloidogenic sequences in between the imperfect repeats which sterically hinder effective chaperone binding.

## 11. Structure-Based Design of Anti-FuBA Peptides

Given that pathological and functional amyloid have different aggregation strategies, do we need to combat their formation in different ways? We can address this question by considering how they respond to specific modulators, such as peptides [87]. Given that amyloid forms through self-recognition, peptides based on the sequence of the target protein are likely to influence aggregation by binding to the protein and thus affect its ability to bind to other copies of the same protein [88,89,90]. This means that the most important step in peptide design is to identify the most aggregation-prone regions (APR) of a given protein. To appreciate this, we need to consider a strategy that has recently emerged from the Schymkowitz-Rousseau SWITCH laboratory, namely, the ability to induce the aggregation of proteins that are otherwise not amyloidogenic (Figure 7).

APR in globular proteins normally hide in the hydrophobic core of the protein where they avoid contact with other proteins; they only become problematic when they are exposed, e.g., during protein biosynthesis where the nascent polypeptide chain emerges from the ribosome and has not yet folded [92,93]. Thus, if a peptide encoding the APR is present in the cell at this stage, it may trap the protein in a misfolded state which can be maintained and used to accumulate additional aggregates by complexing with other copies of the same trapped protein. This strategy has been exploited to induce simultaneous aggregation of many endogenous proteins in a whole range of hosts, ranging from plants and animals to bacteria [91]. The ability to target vital bacterial proteins makes APR peptides potential antibiotics [92,93]. The strategy involves the prediction of APRs in the target protein using computational algorithms [94], followed by the design of peptides containing the APR of the target protein, e.g., tandem repeats of the APR, flanked by charged gatekeeper residues to increase solubility (Figure 7). Such peptides have proven effective in *Staphylococcus epidermidis* and *E. coli* strains [92,93]. We have used the same method to target CsgA in *E. coli*. This may seem pointless given that CsgA is supposed to aggregate anyway, but the advantage of the APR approach is to alter the nature of the ensuing aggregate. Incorporation of peptides into CsgA leads to non-functional aggregates which are less stable than the original CsgA fibrils and reduce biofilm formation [95].

Using a peptide microarray displaying the whole CsgA sequence as staggered arrays of 14-mer peptides, we measured which peptides were particularly good at binding to full-length CsgA. In this way we identified three segments in CsgA which appeared to be the main drivers of CsgA-CsgA interactions: the R3 and terminal R1 and R5 repeats. Additional computational analysis of the aggregation profile of CsgA showed that R1 and R5 contain some of the most aggregation-prone segments. The terminal repeats of CsgA have previously been experimentally verified to form amyloid-like aggregates [96,97], and we therefore decided to focus primarily on APRs within the terminal repeats of CsgA. This is further supported by the predicted β-helical structure of CsgA, which implies that curli amyloid fibrils are formed through stacking of CsgA monomers by intermolecular R1–R5 contacts [45,98,99]. APRs identified in R1 and R5 were then used as the basis for the design of tandem repeat peptides consisting of two identical hepta-sequences (containing the APR identified in either R1 or R5) linked by a flexible (Gly–Ser) or rigid (Pro–Pro) linker. The peptides pushed CsgA to form destabilized aggregates which were less stable towards formic acid and had a different morphology than untreated CsgA according to transmission electron microscopy. We suspect that the peptides work by interacting with monomeric CsgA to promote aggregation into non-native CsgA aggregates.

APRs can be used not only to promote aggregation, but also to block aggregation by blocking the interaction between fibril ends and incoming monomers [100]. This computational design approach uses the APRs of experimentally solved amyloid structures from the PDB as templates; it measures the effect of peptide docking onto the fibril end of the template structures using the FoldX force field [100]. In addition to using an APR amyloid core template from the PDB, the peptide inhibitor calculations can be based on a computationally predicted APR amyloid core template, e.g., predicted from the Cordax method [101]. This approach was extended to the design of structure-based anti-CsgA peptides [95]. Insertions of R and W residues were predicted in silico to have high potential for fibril end-capping [95,100]. We therefore supplemented our anti-CsgA peptide library by designing peptides aimed to “cap” fibril ends by incorporation of Arg and Trp residues into the APR regions of our peptides [100]. Multiple CsgA-targeting peptides worked rapidly on CsgA, inducing a prompt increase in turbidity of the samples [95]. Remarkably, multiple peptides also modulated the fibrillation of FapC, which is possible by binding to FapC segments which are similar to CsgA in terms of sequence and/or structure.

The final proof is an “application-relevant” test, i.e., the impact of the peptides on FuBA fibrillation in a biofilm. Indeed, several peptides reduced biofilm formation in both *E. coli* and *P. aeruginosa*, indicating that our peptides modulated CsgA and FapC fibrillation, in vivo. It is likely that some peptides predominantly function by an “induction mechanism,” where peptides interact with monomeric CsgA and divert the protein into non-native aggregates. This mechanism is supported by the fact that some peptides caused rapid formation of destabilized aggregates with different morphologies, which are much less resistant to formic acid compared to untreated fibrils. Other peptides may work by a “capping mechanism,” where peptides are bound to fibril ends, which blocks further elongation through charge repulsion and steric hindrance. The fibril end “capping” mechanism is supported by our observation that some peptides almost completely suppressed the presence of a sigmoidal thioflavin T fibrillation curve. The peptides designed as fibril end “cappers” appeared to work more efficiently in vitro than the peptides designed to induce non-native aggregation, because they resulted in prompter suppression of the CsgA fibrillation and in higher fibril destabilization towards formic acid. However, both types of anti-CsgA peptide strategies appeared to work well in in vivo anti-biofilm assays. A possible explanation for this could be that the peptides are likely taken up by the bacteria and may therefore work intracellularly. Positively charged residues in peptides have been shown to enhance uptake by bacteria [102]. Our most effective peptides contained 4–9 Arg residues, and the reduction in biofilm formation correlated with the number of Arg residues. Clearly there is scope for more development in this area to target different aspects of bacterial growth and biofilm formation.

## 12. Using Small Molecules and Polyphenols to Target FuBA and Biofilm

Besides smart peptides, another way to target FuBA is to use small molecules acting as aggregation inhibitors. A prime example is the polyphenol epigallocatechin-3-gallate (EGCG) (Figure 8a), which inhibits the formation of both pathological amyloid [103,104,105,106] and FuBA [40]. Although EGCG has shown promising pre-clinical results against pathological amyloids, the compound has not successfully passed clinical trials, possibly because of low chemical stability in vivo (it is prone to epimerization and auto-oxidation above pH 6 [107]) and limited penetration of the blood–brain barrier [106,108]. While EGCG also can inhibit the formation of bacterial amyloid and reduce biofilm formation [40,109], stability issues have also hampered its clinical use against infections [110,111,112]. Nevertheless, EGCG and other (suitably stabilized) small molecules could target biofilm formation in three complementary ways:(i)Biofilm regulation: Impacts on biochemical processes inside or outside the bacterial cell that regulate biofilm formation. For example, EGCG disrupts quorum-sensing (QS) signaling by increasing the binding of pyocyanin (a central QS molecule) to FapC fibrils in *P. aeruginosa* [40]. EGCG also activates expression of the small non-coding RNA molecule RybB that binds to initiation codon of the *csgD* mRNA and inhibits expression of the curli transcription factor CsgD, thereby down-regulating production of the two main components of *E. coli* biofilm, namely, curli and pEtN-cellulose (bacterial cellulose where every second glucose group is modified with phosphoethanolamine) [109,113].(ii)General antibacterial effects: Reduction of biofilm formation at the source, i.e., antimicrobial mechanisms such as disruption of the bacterial cell membrane or inhibition of fatty acid synthesis and enzyme activity [114,115,116].(iii)Direct inhibition: EGCG is known to inhibit fibrillation of amyloidogenic monomers secreted from bacteria and remodeling preformed fibrils to amorphous aggregates [40]. EGCG interacts with FapC monomers and redirects it to relatively stable off-pathway oligomers [117], just as is seen for proteins and peptides involved in neurodegenerative diseases such as α-synuclein and Aβ [118]. As with chaperones [85], the main target of inhibition is the nucleation phase, and FapC/CsgA monomers are redirected to off-pathway oligomers [119]. These off-pathway oligomers are SDS-stable and contain a mixture of β-sheets and random coils [120]. This inhibition of FapC fibrillation can take place even in the presence of amyloid inducers such as SDS, rhamnolipids and LPS [121]. By combining a peptide microarray and the EGCG-binding compound Nitro blue tetrazolium, we have shown that EGCG binds to amyloidogenic hot spots containing the sequence “GVNVAA” in repeats R2 and R3 and even linker regions of FapC sequence [117,122]. Small-angle X-ray scattering measurements revealed a core-shell structure for FapC off-pathway oligomers that consist of ~7 monomers with a 25–26 nm short-axis diameter, which is much bigger than would be expected for on-pathway fibril precursors (~ 10 nm) [117].

There have been many structure–activity studies on EGCG and its sub-scaffolds, such as gallic acid (three hydroxyl groups on a benzoic acid ring), catechin, epicatechin and 3-hydroxytyrosol. However, intact EGCG remains the most effective inhibitor, highlighting the advantages of combining multiple polyphenolic groups. As a logical extension, structures such as penta-*O*-galloyl-β-d-glucose (PGG) that contain five gallic acid groups on a central glucose actually slightly surpass EGCG and show stronger inhibitory effects [117,119,123]. Nevertheless, a fundamental weakness is that the gallic acid groups in both EGCG and PGG are attached to the core of the molecule via ester bonds that are not completely stable above pH 7 [117]. In vitro studies have identified numerous potential alternatives to polyphenols for inhibiting amyloid formation based on in vitro studies [124,125,126], but their use in vivo is a challenge. Fortunately, there is a growing number of studies highlighting the loading of small molecules into nanoparticles, such as lipid vesicles or polymer-coated albumin aggregates [127,128,129,130,131]. This packaging improves the ability to cross the blood–brain barrier (for therapy) or penetrate biofilm (to combat functional amyloid), allowing the small molecules to target amyloidogenic monomers.

Small molecules may also induce the fibrillation of amyloids, as demonstrated with 4,4′-bipyridine (4BPY) (Figure 8b), which induces Aβ fibrillation in vitro [132]. The nitrogen atoms of 4BPY are capable of hydrogen bonding with the -COOH group of the peptides’ C-termini in surface adsorbed peptide assemblies (Figure 8c) [132]. These play a role in our final topic, namely, the ability to assemble bacterial amyloid in an organized fashion on a surface.

## 13. Functional Bacterial Amyloid as a Bionanomaterial

Functional bacterial amyloids are built to last. The fibrillation of CsgA is remarkably robust and can take place under a wide range of conditions, including extreme values of pH [71] and at high denaturant concentrations [63]. Mature CsgA fibrils are also very stable once formed and can withstand high denaturant concentrations and boiling SDS [20,63,133]. In contrast to pathological amyloids, the fibrils formed from CsgA appear to lack polymorphism, suggesting very precise and steady assembly [134]. In addition, CsgA can tolerate quite substantial modifications, e.g., by fusion to large proteins without losing its ability to fibrillate [135,136]. Together, these abilities have motivated the development of several different CsgA-based bionanomaterials [137,138,139,140,141,142]. Among the examples are the fusions of polypeptides which provide completely new functionalities to CsgA, such as increased binding to various types of inorganic substrates, including silica, graphene and silver nanoparticles [136,137,139]. Even more flexibility of CsgA-fusion proteins has been demonstrated using the versatile SpyCatcher-SpyTag technology, which allows a “switch-like” covalent coupling of virtually any other SpyCatcher-tagged heterologous protein to Spy-tagged CsgA [136,137]. Interestingly, even unmodified CsgA can effectively bind many different surfaces, including highly hydrophilic or hydrophobic substrates, and the resulting CsgA protein coatings are very robust against harsh conditions [137,143]. The research of protein interactions with abiotic surfaces is important because as new nanomaterials are being developed, it opens new possibilities for developing novel hybrid bionanomaterials. The effective CsgA fibrillation and the robustness of the final assemblies, combined with CsgA’s high tolerance for protein engineering, makes it a promising candidate for producing novel bionanomaterial [144,145].

## 14. Steering FuBA Assemblies Using the Structured Surface of Graphene

One example of such novel bionanomaterials builds on graphene, a carbon nanomaterial (CN) made of sp [2] bonded carbon atoms connected in a planar hexagonal (honeycomb) lattice [146,147]. Graphene consists of a single planar level of the hexagonal lattice and is one atom thick. Graphene can be produced from graphite, which is simply stacked sheets of graphene. Graphite is both produced by the mining of natural sources and synthesized from high-molecular-weight hydrocarbons [148]. The interactions between amyloids and CN are especially interesting. Both the fibrillation kinetics and the final amyloid assemblies can be drastically modulated by CNs [149]. Interestingly, CN can cause either inhibition or induction of amyloid fibrillation [149]. Spherical fullerenes (also known as C_60_) can bind the hydrophobic motif KLVFF in Aβ and inhibit fibrillation [150]. On the other hand, graphite has been demonstrated to promote the formation of β-strand-dominated assemblies associated with amyloid formation from peptides which are initially dominated by α-helical and random-coil secondary structures [149,151,152,153]. Additionally, the final assemblies of amyloids can be modulated by CN [149]. There is growing evidence that structural patterns in the graphite lattice can be templates for the assembly of amyloids [149,154,155,156]. CsgA aggregation is stimulated by graphite (Figure 9a), and it forms very systematic β-strand rich assemblies on highly oriented pyrolytic graphite (HOPG), manifested as lamellar-like structures using scanning tunneling microscopy (STM) analysis (Figure 9b) [157]. The assemblies are large (>10,000 nm [2]) and do not represent individual fibrils, as observed in other studies, but are instead uninterrupted “film-like” assemblies. The distance between β-strands is ~4.8 Å, and β-strand lengths (~4 nm) are very systematic and correlated over longer distances. Further, the directions of the assemblies appeared to be controlled by the underlying graphite lattice (always with an angle of 60° with respect to each other) (Figure 9c), which led to the expected 4.9 Å separation if the peptides were arranged orthogonally to the sides of the hexagon (Figure 9d). MD simulations have indicated that a CsgA-derived peptide initially adsorbs unstructured clusters which gradually mature into lamellar-like structures in directions guided by the graphite lattice [157].

Finally, the orientation of CsgA peptides can be controlled using the small chaperone-like 4BPY (which hydrogen-bonds to the C-terminus of a peptide sequence [158]), allowing precise and tailored assembly of CsgA peptides on graphite. While individual 4BPY tethering molecules are clearly visible on surfaces assembled with short peptides, they are not seen with full-length CsgA. It is possible that full-length CsgA’s strong preference for ordered assembly (i.e., the avidity effect of having multiple beta-strand motifs available within one polypeptide chain combined with strong van der Waals interactions with graphene) overrules the need for co-assembly with 4BPY. This nicely underlines the strong innate self-organizing principles inherent in functional amyloid.

## Figures and Tables

**Figure 1 molecules-27-04080-f001:**
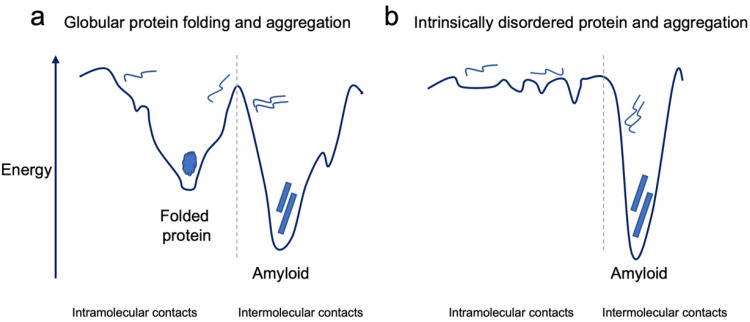
Schematic energy landscapes of globular proteins (**a**) and IDPs (**b**). Both energy landscapes are split into two halves illustrating intramolecular contacts and intermolecular contacts. (**a**) The globular folded protein is the conformation with the lowest energy based on intramolecular contacts. However, aggregation can result in conformations that are even lower in energy, illustrating that amyloid fibril formation is thermodynamically more favorable than the native state. (**b**) IDPs exist in an ensemble of conformations that may have transient intra- and intermolecular contacts. Intermolecular contacts leading to aggregation of IDPs can result in highly stable amyloid aggregates.

**Figure 2 molecules-27-04080-f002:**
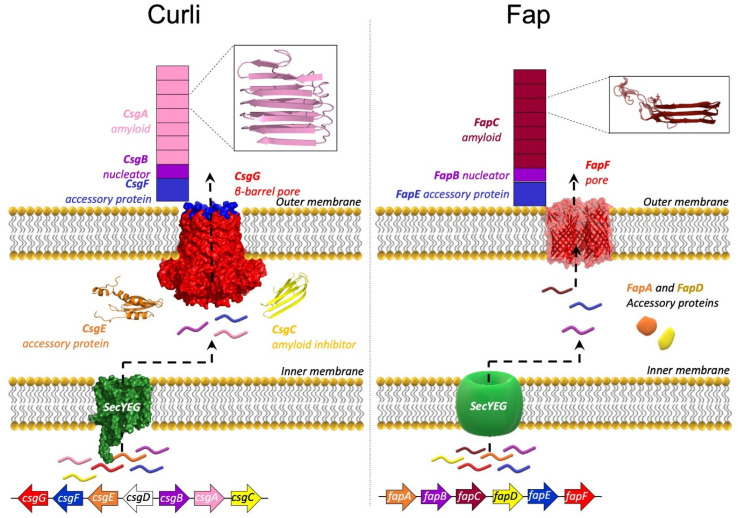
The proteins involved in curli biogenesis are encoded from two divergently transcribed operons, whereas the Fap proteins are encoded from a single operon. The proteins in both systems are transported across the inner membrane through the Sec secretion pathway. The major amyloid components of curli and Fap, CsgA and FapC, respectively, are kept monomeric with the help of chaperone-like accessory proteins, during their transport through the periplasmic space. In each system, the amyloid components are transported to the outer membrane, through their respective pore proteins, CsgG (PDB ID 6SI7) and FapF (PDB ID 5O65). The nucleator proteins CsgB and FapB prime the aggregation of CsgA and FapC, respectively, once the proteins have reached the outer membrane. The insets show structural models of CsgA and FapC. Both CsgA and FapC are predicted to form a β-helix, where the imperfect repeat sequences are stacked on top of each other.

**Figure 3 molecules-27-04080-f003:**
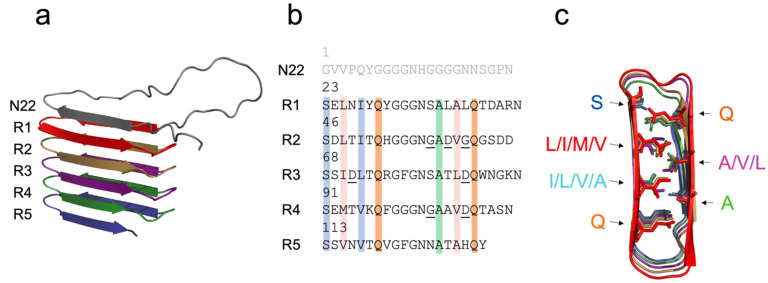
(**a**) CsgA is predicted to form a β-helix (left- or right-handed) where the five imperfect repeats R1–R5 stack on top of each other. Here a right-handed β-helix is shown that was predicted using AlphaFold. (**b**) The primary sequence of CsgA contains an N22 signal sequence, which is important for transport across the outer membrane through CsgG. The five imperfect repeats R1–R5 are found in residues 23–131. The 7 bars illustrate residue ladders formed in the CsgA β-helix due to stacking of conserved residues. Gatekeeper residues (Asp and Gly) are underlined. (**c**) The top-view of the β-helix shown in (**a**). Ladders formed from conserved Ser, Gln, Ala and varied hydrophobic residues are indicated.

**Figure 4 molecules-27-04080-f004:**
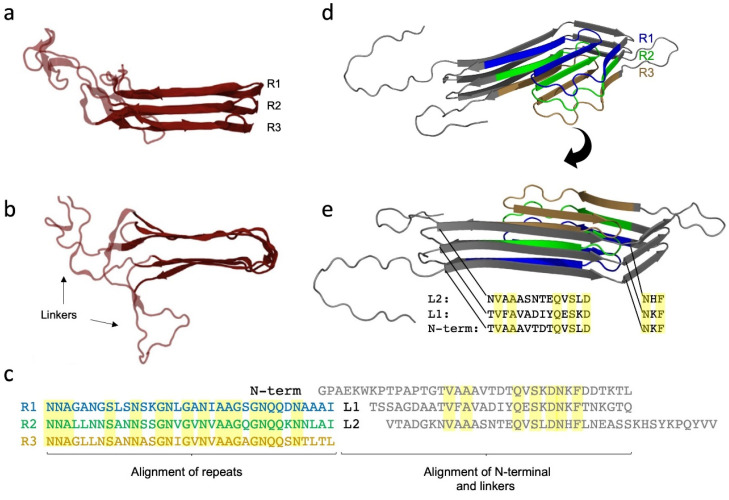
(**a**) FapC is predicted to form a β-helix where the three imperfect repeats R1–R3 stack on top of each other. (**b**) Top-view of the predicted β-helix of FapC. FapC’s long linkers (31 and 39 residue lengths) are marked. Panels a and b modified from [38]. (**c**) Alignment of the primary sequence of the repeats R1, R2 and R3 and the N-terminal and linkers, L1 and L2. “Ladders” formed from different kinds of conserved residues are highlighted. (**d**,**e**) Predicted FapC structure from AlphaFold seen at two different angles. The repeat sequences are colored blue (R1), green (R2), orange (R3).

**Figure 5 molecules-27-04080-f005:**
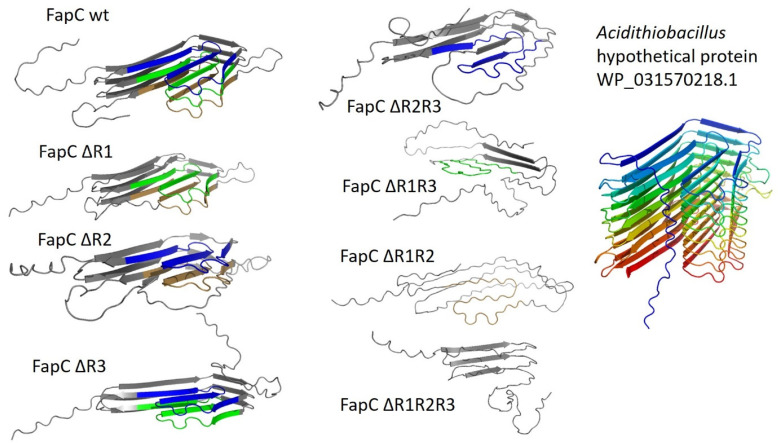
AlphaFold predictions of the structures of FapC imperfect repeats. The two leftmost columns show different FapC UK4 constructs where one or more of the three repeats R1–R3 have been removed (nomenclature and numbering as in [65]), and the structure of the resulting sequence was analyzed by AlphaFold. The rightmost column shows the predicted structure of a FapC homologue from *Acidothiobacillus* which has 8 imperfect repeats, each of which forms a layer in the ensuing β-helix structure. Note the similar cross-sections of FapC wildtype and the *Acidothiobacillus* protein.

**Figure 6 molecules-27-04080-f006:**
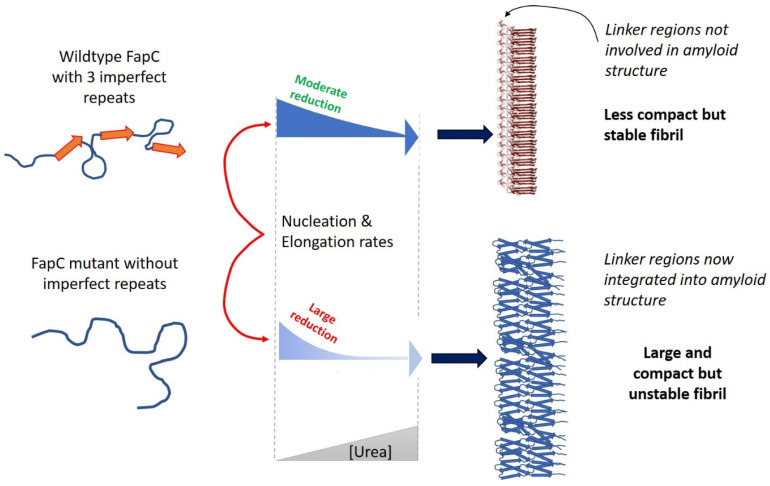
The dependence of FapC fibrillation (nucleation and elongation) on denaturant concentration sheds more light on the folding events accompanying fibrillation. Removal of imperfect repeats leads to a more compact but much less stable fibril, suggesting mobilization of the linker regions to form amyloid. Modified from [63].

**Figure 7 molecules-27-04080-f007:**
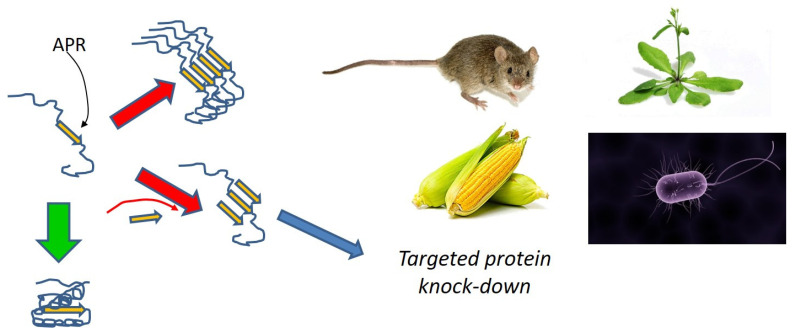
The SWITCH strategy of using APRs (aggregation prone regions) in proteins to knock down protein expression in vivo [91]. APR sequences are usually buried in the protein’s hydrophobic core, but during protein biosynthesis, they may be targeted by peptides containing the same APR sequences, leading to aggregation and (presumably) degradation.

**Figure 8 molecules-27-04080-f008:**
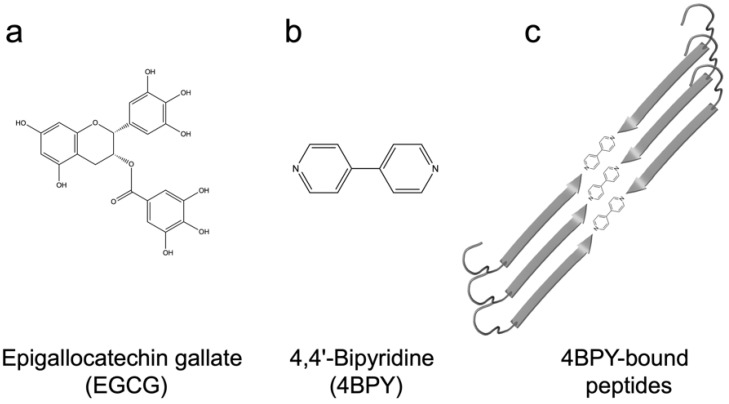
Molecular structures of (**a**) the polyphenol EGCG, (**b**) the chaperone-like small molecule 4BPY. (**c**) 4BPY-bound peptides seen in surface adsorbed assemblies, stabilized by hydrogen bonds between the nitrogen groups of 2BPY and peptide carboxy termini (arrow tips).

**Figure 9 molecules-27-04080-f009:**
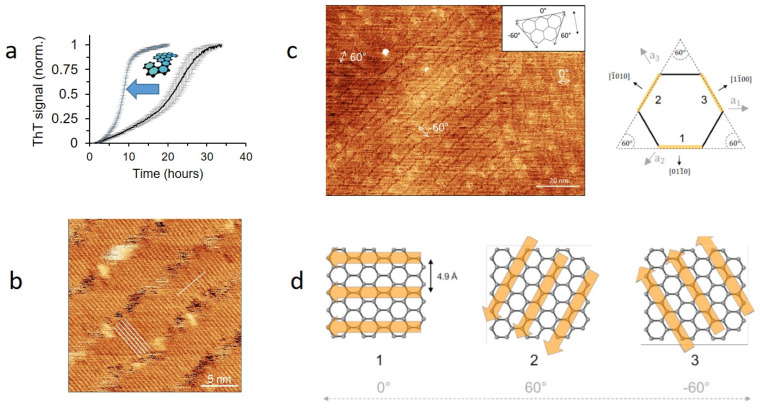
Effect of a structured surface on CsgA self-assembly. (**a**) Graphite nanoparticles promote aggregation of CsgA in vitro. (**b**) STM image of full-length CsgA deposited on HOPG. The five parallel white bars in the lower left corner highlight adjacent β-strands; the orthogonally placed white bar in the upper right corner measures the average distance over 10 strands to be ca. 4.8 Å. (**c**) STM image illustrates the 3 preferred orientations of full-length CsgA on HOPG (angles illustrated in the insert) and their orientation relative to the HOPG orthogonal structure. (**d**) The positions of individual strands in these arrangements are dictated by the orientation of the graphene atoms and lead to a strand distance of 4.9 Å. Adapted from [157].

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
