# Peer review of "Functional Bacterial Amyloids: Understanding Fibrillation, Regulating Biofilm Fibril Formation and Organizing Surface Assemblies"

_molecules, 2022, doi:10.3390/molecules27134080_

Round 1
Reviewer 1 Report
Sønderby TV et al, in the review article, Functional Bacterial Amyloids: Understanding Fibrillation, Regulating Biofilm Fibril Formation and Organizing Surface Assemblies, nicely summarize the characteristics and relevance of different amyloids formed in nature with special emphasis on functional bacterial amyloids (FuBA). In addition to summarizing what is known so far in the field regarding bacterial amyloids, the authors, in several instances, do shed light on important differences between pathological and functional amyloids. The article is very well written and is indeed a true dedication and memorabilia to Christopher Dobson’s works and ideas. The reviewer appreciates the authors for including a section on the potential nanobiotechnological applications of bacterial amyloids. A few suggestions and questions for the authors are listed below. The reviewer believes the comments will be constructive and might help the authors improve the current version of the manuscript.
1) Provide a brief account of the relevance of disordered regions in FuBA and other functional amyloids. Considering consensus sequences demarcating functional and pathological amyloid proteins have not been completely deciphered as yet, it is worth discoursing what causes intrinsically disordered proteins (IDPs) to form proteotoxic aggregates in some cases while not in others.
2) What regulates the magnitude of polymorphism in amyloids? Why do pathological amyloids display a wide variety of heterogeneity compared to functional amyloids? Please offer insights.
3) It is interesting to learn how a human-origin chaperone is directed naturally against bacterial amyloids, whereas a bacterial chaperone affects alpha-synuclein fibrillation. While this property could very well be exploited to design anti-amyloid agents, it is not clear what features (in the chaperones or the target proteins) are responsible for these cross-species interactions, and whether this has any physiological or evolutionary value. It would be nice if the authors include their thoughts and ideas on this.
4) Amyloids formed by bacteria (proteins) and bacterial biofilms have been associated with multiple human diseases. While the SWITCH lab approach with amyloid-prone regions might be helpful given the sequence specificity of the peptides, the use of small molecules or polyphenols does not seem very practical at least in vivo taking into consideration the coexistence of several innocuous mammalian amyloids in the same setting. Authors should highlight this point as a caveat in strategies that rely on small molecule drugs targeting amyloid fibrillation.
5) Please perform a thorough spelling check and ensure the correctness of all references.
Author Response
Response: many thanks for the very constructive feedback which has certainly improved the manuscript.
1) Provide a brief account of the relevance of disordered regions in FuBA and other functional amyloids. Considering consensus sequences demarcating functional and pathological amyloid proteins have not been completely deciphered as yet, it is worth discussing what causes intrinsically disordered proteins (IDPs) to form proteotoxic aggregates in some cases while not in others.
Response: Thank you for helping us to highlight this more clearly. At the end of the section “Amyloids in sickness and in health” we now add “Nevertheless, their formation and structure are highly instructive and provide an excellent showcase of how to control and benefit from a potentially proteotoxic fold. As we will describe in greater detail below, the best-studied functional amyloids (those in E. coli and Pseudomonas) are organized as multiple copies of relatively simple repeats which lack structure in the monomeric state but fold at the multimeric level in a simple but robust framework thanks to a large number of stabilizing inter- and intramolecular contacts which are built up in a hierarchical manner. This is in stark contrast to pathological amyloid proteins where there are no repeat sequences to provide an orderly framework for assembly; consequently many different aggregate types (polymorphs) exist which can form in different ways and typically involve cytotoxic partially folded intermediate states.”
2) What regulates the magnitude of polymorphism in amyloids? Why do pathological amyloids display a wide variety of heterogeneity compared to functional amyloids? Please offer insights.
Response: We hope that our response to the previous question addresses this.
3) It is interesting to learn how a human-origin chaperone is directed naturally against bacterial amyloids, whereas a bacterial chaperone affects alpha-synuclein fibrillation. While this property could very well be exploited to design anti-amyloid agents, it is not clear what features (in the chaperones or the target proteins) are responsible for these cross-species interactions, and whether this has any physiological or evolutionary value. It would be nice if the authors include their thoughts and ideas on this.
Response: We have some text in red to the section on chaperones as follows: “At the same time, the fact that so many different chaperones with no evolutionary relationship to each other can target the same (optimized) aggregators suggest a commonality of anti-amyloid activities across Nature, perhaps driven by the recognition of amyloidogenic hotspots within the primary structure. In support of this, CsgC can inhibit human αSN fibrillation 1, 2 and reduces both CsgA and FapC fibrillation in vitro 3. The specific molecular basis for chaperone action must remain speculative, though the efficiency of inhibition seems to scale with affinity of binding to the monomeric state4. In addition, we note that FapC is more resistant to chaperone action than CsgA and this may reflect the presence of less amyloidogenic sequences in between the imperfect repeats which sterically hinder effective chaperone binding.”
4) Amyloids formed by bacteria (proteins) and bacterial biofilms have been associated with multiple human diseases. While the SWITCH lab approach with amyloid-prone regions might be helpful given the sequence specificity of the peptides, the use of small molecules or polyphenols does not seem very practical at least in vivo taking into consideration the coexistence of several innocuous mammalian amyloids in the same setting. Authors should highlight this point as a caveat in strategies that rely on small molecule drugs targeting amyloid fibrillation.
Response: This is a fine point. At the end of the section on EGCG we add: “In vitro studies have identified numerous potential alternatives to polyphenols in inhibiting amyloid formation based on in vitro studies 5-8 but their use in vivo is a challenge. Fortunately there is an growing number of studies highlighting the loading of small molecules into nanoparticles such as lipid vesicles or polymer-coated albumin aggregates 9-13. This packaging improves the ability to cross the blood-brain barrier (for therapy) or penetrate biofilm (to combat functional amyloid), allowing the small molecules to target amyloidogenic monomers. “
We appreciate the reviewer’s point about unwanted side-effects against human functional amyloid, however the limited insight into e.g. targeting of pmel17 (probably the most well-studied human functional amyloid) makes it rather speculative for us to theorize about this. Furthermore, we focus on bacterial amyloid where collateral damage is an advantage.
5) Please perform a thorough spelling check and ensure the correctness of all references.
Response: This has been done.
Reviewer 2 Report
In this work the authors describe and analyze the Functional Bacterial Amyloids (FuBA), especifically in Escherichia coli and Pseudomonas aeruginosa, presenting curli and Fap systems, and their major amyloid components, CsgA and FapC, respectively.
They present the processes of fibrillation, the design of anti-FuBA peptides and the proposal of FuBA as a bio-nano-material.
In my opinion, the manuscript is clearly written, presented correctly and the proposed questions addressed with an interesting approach, also giving a fluid reading.
I only have really minor comments:
* The different described techniques, as for example Molecular Dynamics Simulations should be written in capital letters.
* Line 200 - The word electron is repeated.
* To make the descriptions clearer, I suggest not using in the same sentence 1-letter and 3-letter codes for aminoacids, as in lines 228-230 and 463-465.
* Line 237 - Where it says "show" it should say "shown".
* Line 260 - There is an "is" out of place.
* Line 268 - There is a missing "in" between "and" and "the".
* Line 288 - "form" instead of "forms"?
* Line 475 - It says "fct" instead of "fact".
* Lines 575-576 - "bio-nano-materials" instead of "bio-nano-material"?
* Line 581 - There is an extra "e" in the word "simplye".
Author Response
Response: Thank you very much for these points which had cleared eluded our attention. Regarding “bio-nano” we can see from the literature that it is most widespread to avoid hyphens, so we now simply use “bionanomaterials” etc. throughout.
Reviewer 3 Report
The review manuscript was submitted to a special issue in memory of Professor Sir Christopher M. Dobson, a pioneer and expert of the field amyloid proteins. The authors are active researchers in the field with a lot of good works, and are thus proper candidates to write such a review. The manuscript was well written, providing insightful convents on various aspects of amyloid proteins. I have only some minor comments:
(1) Lines 21, 493: "Smart peptides" should be "Small peptides"?
(2) Line 53: "at my then host institute"? I am not sure whether that is proper in English. (I am not a native speaker of English.)
(3) On Fig.1: If a protein has a lower stability in the folded state, it would likely have a lower stability in the amyloid form, since folded state and amyloid state utilize similar driving forces such as hydrophobic interaction, hydrogen bonding, etc. Therefore, in Fig. 1(b), when the free energy of the folded state gets higher, that of the amyloid state hardly keeps unchanged.
(4) It is noted that the amyloid propensity is well correlated with the propensity of intrinsically disordered proteins. For example, F. Jin, et al, Inherent relationships among different biophysical prediction methods for intrinsically disordered proteins. Biophys. J. 104, 488 (2013).
Author Response
Response: Many thanks for these useful and encouraging comments.
(1) Lines 21, 493: "Smart peptides" should be "Small peptides"?
Response: Actually we meant smart peptides since they were designed using (hopefully) smart principles.
(2) Line 53: "at my then host institute"? I am not sure whether that is proper in English. (I am not a native speaker of English.)
Response: We agree this is a clumsy expression. We have changed to “at my host institute (at the time Aalborg University”)
(3) On Fig.1: If a protein has a lower stability in the folded state, it would likely have a lower stability in the amyloid form, since folded state and amyloid state utilize similar driving forces such as hydrophobic interaction, hydrogen bonding, etc. Therefore, in Fig. 1(b), when the free energy of the folded state gets higher, that of the amyloid state hardly keeps unchanged.
Response: The reviewer brings up an interesting point. However, we would argue that just because the protein is an IDP in the monomeric state, that does not necessarily destabilize the amyloid state – this is because the folding patterns are not the same in the native monomer and the amyloid. FapC and CsgA are excellent examples of IDPs which are converted into super-stable amyloids upon amyloid formation 14, 15.
(4) It is noted that the amyloid propensity is well correlated with the propensity of intrinsically disordered proteins. For example, F. Jin, et al, Inherent relationships among different biophysical prediction methods for intrinsically disordered proteins. Biophys. J. 104, 488 (2013).
Response: This is an excellent point. In fact, it fits perfectly with our own observations e.g. for FapC, where our analyses showed that the repeat sequences (R1, R2 and R3) are also predicted to have the highest degree of intrinsic disorder15.
We have added these considerations to our discussion with the paragraph (page 13) in which we refer to Jin et al::
“Interestingly, in computational disorder predictions of FapC, the repeat sequences obtained higher scores of disorder propensity compared to the linker regions 16. This is in good agreement with previous systematic analysis which showed a clear relationship between intrinsic disorder and amyloid propensity 17”